# Median Selection with Noisy and Structural Information

**Chenglin Fan**[*]
Seoul National University
Seoul 08826, South Korea
fanchenglin@snu.ac.kr

**Mingyu Kang**[*]
Seoul National University
Seoul 08826, South Korea
mgmh26@snu.ac.kr

## Abstract

We study the problem of computing the exact median by leveraging side information to minimize costly, exact comparisons. We analyze this problem in two key settings: (1) using predictions from unreliable "weak" oracles, and (2) exploiting known structural information in the form of a partial order. In the classical setting, we introduce a modified LazySelect algorithm that combines weak comparisons with occasional strong comparisons through majority voting. We show that this hybrid strategy has near-linear running time and can achieve high-probability correctness using only sublinear strong comparisons, even when the weak oracle is only slightly better than random guessing. Our theoretical results hold under the *persistent comparison model*, where resampling will not amplify the probability of correctness. In the partially ordered setting, we generalize the notion of median to directed acyclic graphs (DAGs) and show that the complexity of median selection depends heavily on the DAG's width. We complement our analysis with extensive experiments on synthetic data.

## 1 Introduction

Median selection is a fundamental problem in algorithm design, with wide-ranging applications in machine learning, data analysis, and decision-making. Given a set of $n$ elements with an unknown total order, the goal is to identify the median—the element of rank $\lfloor n/2 \rfloor$. Finding the median of a set is a classical problem in computer science and statistics, with applications ranging from robust statistics to machine learning and decision-making. The traditional model assumes that comparisons are perfect and equally costly, but in many modern settings—such as crowdsourcing [Karger et al., 2011], preference elicitation [Lu and Boutilier, 2011], or federated analytics [Kairouz et al., 2021]—comparisons can be noisy, expensive, or even unreliable.

Our work addresses these challenges through a unified framework in which side information is leveraged to reduce reliance on expensive, definitive comparisons. We consider two forms of side information: predictions from a cheap but noisy "weak" oracle, and structural knowledge in the form of a pre-existing partial order. The latter can be viewed as the outcomes of trusted, historical comparisons, framing both settings as instances of a common problem: how to efficiently compute the median when partial or imperfect comparison information is already available. This setting is motivated by recent work in *learning-augmented algorithms* [Lykouris and Vassilvitskii, 2021, Purohit et al., 2018], which aim to bridge the gap between traditional worst-case algorithms and data-driven approaches.

A key insight is that in many practical scenarios, the elements to be compared may already have some latent structure. For example, in partially ordered sets or when the input items are embedded

---

[*]Authors listed alphabetically. All authors contributed equally to this work.

39th Conference on Neural Information Processing Systems (NeurIPS 2025).

in a known hierarchy, some comparisons may be inferred or ruled out. Exploiting such structural properties has proven useful in sorting and selection [Daskalakis et al., 2011].

We formalize this intuition and present two main algorithmic results: (1) when weak comparisons are available with some bias toward the correct order, we show how to combine them with a small number of strong comparisons to recover the median; (2) when the input set has a known partial order (e.g., a DAG), we show how to use the structure to *reduce the number of strong comparisons to sublinear*, while ensuring correctness. Our results are accompanied by theoretical bounds and suggest a new paradigm for combining *structural priors* and *learning predictions* in classic decision problems.

## 1.1 Related Works

### 1.1.1 State-of-the-Art Algorithms for Finding the Median

There exist two main classes of median selection algorithms: **randomized algorithms** and **deterministic algorithms**.

**Randomized Algorithms**

- **QuickSelect**: A randomized variant of QuickSort [Hoare, 1961], QuickSelect randomly selects a pivot and partitions the array, recursively narrowing down to the median. Its expected time complexity is $O(n)$, but in the worst case, it can be $O(n^2)$ with unlucky pivots.
- **LazySelect**: Known as Rivest and Floyd's Select algorithm [Floyd and Rivest, 1975], LazySelect samples $r$ elements, selects two pivots from the sorted sample, and partitions the array into three regions. It sorts the middle region and returns the median. It runs in expected linear time with high probability, achieving $O(n)$ with probability $1 - O(n^{-1/4})$.

**Deterministic Algorithms**

- **Sorting-Based Approach**: A simple deterministic method involves sorting the array (e.g., MergeSort, HeapSort) [Cormen et al., 2022] and selecting the middle element. Its worst-case complexity is $O(n \log n)$, making it inefficient for large datasets.
- **Median of Medians** [Blum et al., 1973]: To improve QuickSelect's worst-case performance, this algorithm deterministically selects a pivot in $O(n)$ time, ensuring a worst-case complexity of $O(n)$.

### 1.1.2 Learning-Augmented Algorithms

Recent advancements in machine learning have inspired learning-augmented algorithms, which leverage additional probabilistic information to enhance traditional algorithms. Learning-augmented algorithms do not rely solely on standard comparisons but instead incorporate predictions that guide the selection process. This paradigm is particularly useful when comparisons have different costs or reliability levels, as in the case of strong and weak comparison oracles.

Learning-augmented algorithms have been successfully applied in various domains, including:

- **Algorithms with Predictions**. While much of the prior research focuses on online settings such as caching [Lykouris and Vassilvitskii, 2021, Rohatgi, 2020, Bansal et al., 2022], rent-or-buy problems [Purohit et al., 2018, Angelopoulos et al., 2024], and scheduling [Azar et al., 2021, Lindermayr and Megow, 2022], recent advancements have extended these techniques to offline problems. Notable examples include matching [Dinitz et al., 2021], clustering [Ergun et al., 2021], and sorting [Bai and Coester, 2023, Lu et al., 2021]. Inspired by Kraska et al. [2018], several studies also examine data structures [Lin et al., 2022] and robust use of multiple predictors [Anand et al., 2022, Antoniadis et al., 2023]. Recent works continue to broaden the field's reach: binary search with distributional predictions [Dinitz et al., 2024], competitive warm-start strategies for algorithms with predictions [Blum and Srinivas, 2025], dynamic graph algorithms with predictions [Brand et al., 2024], and learning-augmented priority queues [Benomar and Coester, 2024] highlight the versatility of predictions in diverse algorithmic domains.

- **Sorting with Prediction**. Lu et al. [2021] studied generalized sorting with forbidden comparisons and prediction errors similar to our weak comparisons, achieving bounds of $O(n \log n + w)$ and $O(nw)$, though the latter degrades to $O(n^2)$ with one error per item—ours remains $O(n)$. Erlebach et al. [2023] explored sorting and hypergraph orientation under uncertainty with predictions, aiming to minimize the number of queries. Bai and Coester [2023] proposed learning-augmented sorting algorithms using predicted positions and fast-but-noisy comparisons, achieving optimal $O\left(\sum_i \log \eta_i\right)$ exact comparisons ($\eta_i$ denote the prediction error of the rank of $i$-th element), which degrades smoothly from $O(n)$ to $O(n \log n)$ as prediction error increases.

### 1.1.3 Sorting with Structural and Noisy Information

Many real-world applications require sorting in the presence of uncertainty or pre-existing structure. In the *noisy comparison model*, comparison outcomes may be unreliable. The *independent noisy model* assumes each comparison is flipped independently with probability $p \in (0, 1)$, where $p$ is the error probability. Recent work by Gu and Xu [2023] established optimal bounds for sorting $n$ elements under this model. In the *recurrent noisy setting*, proposed by Braverman and Mossel [2007], repeated queries between the same pair return consistent outcomes. Geissmann et al. [2018] gave an optimal algorithm in this setting with $O(n \log n)$ time, $O(\log n)$ maximum dislocation, and $O(n)$ total dislocation with high probability. These approaches generally focus on approximate sorting, and to the best of our knowledge, no prior work considers combining both clean (reliable) and dirty (unreliable) comparisons in exact sorting.

In parallel, *adaptive sorting* methods aim to exploit partial structure in the input to reduce computational cost. Examples include TimSort [Peters, 2002] and Powersort [Munro and Wild, 2018]. These algorithms take advantage of presortedness, typically measured via global metrics like the number of inversions or runs. Estivill-Castro and Wood [1992] provide a comprehensive overview. While related in spirit, traditional adaptive sorting methods do not utilize explicit predictions or distinguish between comparison quality. Furthermore, their performance bounds are often based on aggregate error measures rather than element-wise guarantees.

## 1.2 Our Results

In this paper, we revisit median selection through two modern lenses that capture realistic constraints in learning and decision-making systems:

**Median Selection with Strong and Weak Oracles.** We formalize this setting using two comparison oracles: a strong (accurate but expensive) oracle and a weak (cheap but noisy) oracle. Our modified LAZYSELECT algorithm strategically trades a higher total number of oracle calls for a drastic reduction in costly strong queries. While the original LAZYSELECT requires $O(n)$ strong comparisons in expectation, our method reduces this to $\tilde{O}(n^{3/4})$ strong comparisons, at the cost of $O(n \log n)$ weak ones. This trade-off is highly advantageous in settings where strong queries are orders of magnitude more expensive.

**Theorem 1** (Main Result 1). *Let $A$ be an array of $n$ elements and assume the weak comparison oracle errors independently with probability $p < 1/2$. Then, with high probability, the modified LazySelect algorithm returns the median of $A$ using (1) $O(n \log n)$ running time, (2) $O(n \log n)$ weak comparisons, (3) $\tilde{O}(n^{3/4})$ strong comparisons, (4) and succeeds with probability at least $1 - O(n^{-1/4})$.*

**Structure-Aware Median Selection in Partially Ordered Sets.** In structured domains, partial orderings among elements are often known *a priori*, represented as a directed acyclic graph (DAG). Such structure arises naturally in scheduling, causal inference, and preference modeling. We study median selection in this setting and design an efficient algorithm that exploits the DAG structure to minimize the number of necessary comparisons. Our approach prunes redundant queries and guides the search using order-theoretic insights.

**Theorem 2** (Main Result 2). *Let $G = (V, E)$ be a DAG over $n$ elements with width $w$, and let the median element be selected using Algorithm 3. Then the expected number of total comparison cost:*

$$O\left(\left(\log n + w \log\left(\frac{n}{w}\right)\right) \log n\right).$$

*In particular:*

- *If $w = O(1)$, the total number of comparisons is sublinear in $n$, specifically $O(\log^2 n)$. In the special case where the DAG is a single chain $w = 1$), the recursion terminates after one step, and the median is found in $O(\log n)$ comparisons.*

- *The comparison cost is $o(n)$ whenever $w \log(n/w) = o(n/\log n)$, outperforming classical $\Theta(n)$ methods for a wide range of sparse DAGs.*

Our results provide new algorithmic techniques for median selection in realistic, structured, and learning-augmented environments. By leveraging side information—whether in the form of approximate predictions or partial orders—we demonstrate that substantial improvements over classical baselines are possible. Our work contributes to the growing body of research at the intersection of algorithms, learning theory, and practical decision systems.

## 2 Preliminary

### 2.1 Median Selection with Strong and Weak Oracles

We are given a set $S = x_1, x_2, \ldots, x_n$ of $n$ distinct elements, and the task is to identify the median element—i.e., the element that ranks $\lfloor n/2 \rfloor$-th in the total order induced by their true values.

To perform comparisons, we have access to two types of oracles:

- **Strong Comparison Oracle** $\mathcal{O}_s(a, b)$: Given any two elements $a, b \in S$, this oracle returns the correct result of the comparison—whether $a < b$ or $a > b$—with certainty. However, it is expensive and should be used sparingly.

- **Weak Comparison Oracle with Persistent Errors** $\mathcal{O}_w(a, b)$: Given any two elements $a, b \in S$, this oracle returns a possibly noisy answer whether $a \overset{.}{<} b$ or $a \overset{.}{>} b$: it returns the correct comparison outcome with probability $1 - p > 1/2$, and the incorrect one with probability $p$. Once the value of $\mathcal{O}_w(a, b)$ is fixed, it is used consistently throughout the execution (i.e., the error is persistent, resampling will not amplify the probability of correctness under this model).

**Goal.** The objective is to *identify the true median* of a set of $n$ elements using a combination of strong and weak comparison oracles, such that:

1. **Correctness Guarantee:** The algorithm returns the exact median element $x^* \in S$ (i.e., the $\lfloor n/2 \rfloor$-th smallest element in the true total order) *with high probability*, typically at least $1 - \delta$, for some small $\delta > 0$.

2. **Efficient Use of Oracles:**
    - The *expected number of queries to the strong oracle* is *sublinear in $n$*—e.g., $o(n)$ or $\tilde{O}(n^{3/4}, \sqrt{n})$—to reduce reliance on costly but accurate comparisons.
    - The *number of queries to the weak oracle* is also *minimized*, ideally remaining near-linear or better, while ensuring sufficient confidence to guide the use of strong oracle queries.
    - The total running time should be $\tilde{O}(n)$.

3. **Robustness to Noise:** The algorithm must tolerate uncertainty in the weak oracle (which returns correct outcomes with probability $1 - p > 1/2$) and strategically integrate both oracles to ensure reliable decision-making.

### 2.2 Median Selection with Partial Order Prediction

Given a set of $n$ elements with an associated partial order represented as a directed acyclic graph (DAG) $G = (V, E)$, the objective is to identify the *median element*—that is, an element $x^* \in V$ such that at most $\lfloor n/2 \rfloor$ elements precede $x^*$ and at most $\lfloor n/2 \rfloor$ elements succeed $x^*$ in the true (but initially unknown) total order.

Formally, assuming the true total order $\prec$ is consistent with the DAG $G$ (up to potential noise), we seek an element $x^*$ such that:

$$|\{v \in V : v \prec x^*\}| \leq \left\lfloor \frac{n}{2} \right\rfloor \quad \text{and} \quad |\{v \in V : v \succ x^*\}| \leq \left\lfloor \frac{n}{2} \right\rfloor.$$

The DAG $G$ provides incomplete pairwise precedence information, which may be derived from prior knowledge, historical comparisons, or predictive models. Our algorithm may query pairwise comparisons directly (e.g., via a comparison oracle) to resolve uncertainty. The goal is to exploit the DAG structure to (1) minimize the number of explicit comparisons required to find the median, and (2) avoid redundant comparisons by leveraging the transitive closure within the DAG.

The efficiency of our structure-aware algorithm depends on the width of the DAG, which is formally defined as the size of the largest antichain (a subset of vertices where no two are comparable under the partial order).

# 3 Median Selection with Strong and Weak Oracle

This section introduces LazySelect with Majority-Vote Weak Oracle, a randomized algorithm for computing the median of an array using a combination of weak and strong comparison oracles.

## 3.1 Algorithm

We present a robust adaptation of the LazySelect algorithm that operates with a *weak comparison oracle*, where each pairwise comparison returns an incorrect result with probability $p < \frac{1}{2}$. To mitigate the effect of this noise, we compare each input element against a set of reference elements drawn from the sampled range boundaries and apply a **2/3-majority rule** over the aggregate outcomes. This ensures that the overall algorithm remains robust while preserving its asymptotic performance. The algorithm is detailed in Algorithm 1.

**High-Level Idea:** The algorithm aims to efficiently compute the median of an array using unreliable (weak) comparisons, which may return incorrect results with some probability. To overcome this, it first selects and sorts a small random sample from the input to estimate a central range $[d, u]$ likely to contain the true median. Each input element is then compared against a small set of reference elements near the boundaries of this range. Based on the majority of these weak comparisons, elements are classified as likely smaller, larger, or ambiguous. Only ambiguous elements are resolved using a reliable (strong) oracle. If this classification isolates a small set that must contain the median, the algorithm sorts this set and directly extracts the median. Otherwise, it reports failure with low probability.

The following steps detail how this idea is implemented in practice.

**Algorithm Description**: The algorithm proceeds as follows:

1. Let $r = n^{3/4}$, $k = \sqrt{n}$. Sample $r$ elements uniformly at random from the input array $A$ to form a subset $S$, and sort $S$.

2. Define lower and upper bounds as $d = S[\frac{r}{2} - k]$ and $u = S[\frac{r}{2} + k]$.

3. Let $D$ be the largest $c \log n$ elements in $\{x \in S \mid x \leq d\}$, and $U$ be the smallest $c \log n$ elements in $\{x \in S \mid x \geq u\}$.

4. For each $a \in A$, compare $a$ to each element of $D \cup U$ using weak comparisons and record the number of times the result is $<$ (denoted count$_<$) and $>$ (denoted count$_>$).

5. If count$_< > \frac{2}{3} \cdot$ (count$_<$ + count$_>$), assign $a \in L$. If count$_> > \frac{2}{3} \cdot$ (count$_<$ + count$_>$), assign $a \in R$. Otherwise, use a strong oracle to classify $a$ into $L$, $M$, or $R$.

6. If classification succeeds with $|L| < n/2 < |L| + |M|$, sort $M$ and return the $(n/2 - |L| + 1)$th element of $M$ as the median.

7. If the classification fails, report an error.

---

**Algorithm 1** LazySelect with Majority-Vote Weak Oracle

---

**Input:** array $A$ of size $n$
**Output:** Median element of $A$

1   $r \leftarrow n^{3/4}$, $k \leftarrow \sqrt{n}$
2   Sample $r$ elements from $A$ to form $S$, and sort $S$          ▷ strong oracle
3   $d \leftarrow S[\frac{r}{2} - k]$, $u \leftarrow S[\frac{r}{2} + k]$
4   $D \leftarrow$ largest $c \log n$ elements $\leq d$, $U \leftarrow$ smallest $c \log n$ elements $\geq u$
5   Initialize $L$, $R$, and $M$ as empty
6   **for** each $a \in A$ **do**
7      $\text{count}_<, \text{count}_> \leftarrow 0$
8      **for** each $d' \in D$, $u' \in U$ **do**
9         Compare $a$ to $d'$ and $u'$ and update $\text{count}_<$, $\text{count}_>$      ▷ weak oracle
10     $\text{total} \leftarrow \text{count}_< + \text{count}_>$
11     **if** $\text{count}_< > \frac{2}{3} \cdot \text{total}$ **then** $L \leftarrow L \cup \{a\}$
12     **else if** $\text{count}_> > \frac{2}{3} \cdot \text{total}$ **then** $R \leftarrow R \cup \{a\}$
13     **else** Compare $a$ with $d$ and $u$ to decide whether to place $a$ into $L$, $M$, or $R$.   ▷ strong oracle
14   **if** classification successful **then**
15     Sort $M$ and return $M[\frac{n}{2} - |L| + 1]$
16   **else**
17     **return** error

---

## 3.2   Analysis

This lemma formalizes the idea that the "middle" set $M$ remains small. Intuitively, since our pivots $d$ and $u$ are chosen from a large random sample, the number of elements from the full array that truly fall between them is small. Furthermore, our majority-vote scheme makes misclassification into $M$ a rare event, ensuring that the total size of $M$ stays bounded.

**Lemma 3** (Size of the Middle Region). *With probability $1 - O(n^{-1/4})$, the number of elements in the middle set $M$ is $O(n^{3/4})$.*

Here, we quantify the robustness of the majority vote. By performing $O(\log n)$ weak comparisons, we leverage concentration bounds (such as Hoeffding's inequality). Since the oracle is better than random, the probability that the majority vote is incorrect decays exponentially, making misclassification polynomially unlikely in $n$.

**Lemma 4** (Misclassification Probability). *For an element truly in $L$ (respectively $R$), the probability of misclassification into $M$ is $O(n^{-\gamma})$ for some constant $\gamma = \Theta(1)$, provided $p < 1/2$.*

This complexity bound follows directly from the previous lemmas. The costly strong oracle is invoked only for elements that fall into the middle set $M$. Since Lemma 5 guarantees that $|M| = O(n^{3/4})$ with high probability, the cost of sorting this set remains sublinear.

**Lemma 5** (Strong Comparison Complexity). *With probability $1 - O(n^{-1/4})$, the algorithm uses $\tilde{O}(n^{3/4})$ strong oracle queries.*

The proofs of Lemmas 3, 4, and 5 are deferred to the Appendix.

We now summarize the full performance and correctness guarantees of the algorithm:

**Theorem 6.** *Let $A$ be an array of $n$ elements, and suppose the weak comparison oracle returns an incorrect result independently with probability $p < \frac{1}{2}$. Then, with high probability, the modified LazySelect algorithm returns the median of $A$ with the following guarantees: $O(n \log n)$ total running time, $O(n \log n)$ weak comparisons, $\tilde{O}(n^{3/4})$ strong comparisons, and success probability at least $1 - O(n^{-1/4})$.*

## 3.3   Generalization to the $k$-th Largest Element

Algorithm 1 extends naturally to finding the $k$-th largest (or smallest) element by adjusting pivot selection and the success condition while preserving the core idea of using a random sample to narrow

the search. Given a sample $S$ of size $r$, pivots are chosen to bracket the expected rank of the $k$-th element, defined as

$$d = S\left[\left\lfloor r\frac{k}{n} - c_k \right\rfloor\right], \quad u = S\left[\left\lceil r\frac{k}{n} + c_k \right\rceil\right],$$

where $c_k$ ( we choose $c_k = \sqrt{n}$ as before) is a cushion parameter ensuring, with high probability, that the true $k$-th element lies in $[d, u]$. The success condition becomes $|L| < k \le |L| + |M|$, where $L$ and $M$ are elements below $d$ and between $d, u$ respectively; when satisfied, $M$ is sorted and $M[k - |L|]$ returned. These modifications preserve sublinear strong comparisons and high success probability, enabling efficient retrieval of any $k$-th order statistic.

**Remark.** *Achieving high-probability median selection using only $O(n)$* weak queries *(with correctness probability $p = \frac{1}{2} + \epsilon$) and $o(n)$ strong queries is fundamentally challenging. This follows from two key barriers. First, the* weak oracle limitation*: prior work Braverman and Mossel [2007] shows that $O(n)$ weak queries misclassify each element with constant probability, producing $\Theta(n)$ uncorrected errors. Second, the* strong oracle sparsity*: with only $o(n)$ strong queries, at most $o(n)$ of these errors can be corrected. As a result, $\Theta(n) - o(n) = \Theta(n)$ errors remain, leading to median misidentification with probability $1 - o(1)$. This highlights a fundamental trade-off between query quality and quantity in median selection.*

# 4 Median Selection with Structural Information

In this section, we develop a median selection algorithm that exploits known partial order information represented by a directed acyclic graph (DAG). The key insight is that when a partial order is available, many pairwise comparisons used in classical algorithms can be avoided. Our proposed algorithm leverages structure in the DAG to reduce the number of comparisons while still guaranteeing correctness. We present formal analysis establishing the correctness and efficiency of our method, and show how partitioning and refinement operations can be conducted efficiently using graph traversal and chain decomposition techniques.

## 4.1 Motivation and Overview

Let $A = \{a_1, \ldots, a_n\}$ be a set of $n$ elements, and let $G = (V, E)$ be a DAG over the elements in $A$, where $(a_i, a_j) \in E$ implies $a_i \prec a_j$. The DAG encodes partial information about the relative order of the elements. Our objective is to find the *true median* of $A$, defined as an element $x^* \in A$ such that at most $\lfloor \frac{n}{2} \rfloor$ elements precede $x^*$ and at most $\lfloor \frac{n}{2} \rfloor$ elements succeed $x^*$ in the total order, using as few comparisons as possible.

The central idea is to use the DAG structure to predict the position of elements relative to a chosen pivot without performing explicit comparisons. We decompose the DAG into disjoint chains (by Dilworth's theorem), and refine partitions via binary search within each chain.

## 4.2 Algorithm Description

The algorithm performs median selection in a partially ordered set represented as a DAG by recursively partitioning the graph around a pivot. The PartitionWithDAG procedure uses BFS and chain decomposition (via Dilworth's Theorem) to determine which elements are less than or greater than the pivot. The MedianSelectionUsingDAG algorithm then recursively searches in the left or right subset depending on the size of the partition until it finds the median. The general procedure for partitioning using a DAG structure is outlined in Algorithm 2, while the randomized median selection using a DAG is presented in Algorithm 3, and the deterministic median selection using a DAG is detailed in Algorithm 4 in the Appendix.

---

**Algorithm 2** Partition with DAG Structure

---

**Input:** DAG $G = (V, E)$, pivot element $q \in V$
**Output:** Sets $L$ and $R$, where $L = \{x \in V \mid x \prec q\}$, $R = \{x \in V \mid x \succ q\}$
  1  Initialize $L \leftarrow \emptyset$, $R \leftarrow \emptyset$
  2  Perform BFS from $q$ in the reverse direction to find all $x$ with a path $x \to \cdots \to q$; add to $L$
  3  Perform BFS from $q$ in the forward direction to find all $x$ with a path $q \to \cdots \to x$; add to $R$
  4  Decompose $G$ into chains $C_1, \ldots, C_w$ using Dilworth's Theorem
  5  **for** each chain $C_i$ **do**
  6      **if** $q \in C_i$ **then**
  7          Use binary search in $C_i$ to classify remaining elements relative to $q$
  8      **else**
  9          Use binary search in $C_i$ to determine if any elements are comparable to $q$
10          Add elements smaller/larger than $q$ to $L/R$ accordingly
11  **return** $L, R$

---

---

**Algorithm 3** Randomized Median Selection Algorithm Using DAG

---

**Input:** DAG $G = (V, E)$, where $V$ is a set of $n$ elements
**Output:** Median element of $V$
  1  **if** $n = 1$ **then**
  2      **return** the single element
  3  Select a pivot $q \in V$ (randomly or deterministically)
  4  $L, R \leftarrow$ PARTITIONWITHDAG$(G, q)$
  5  **if** $|L| = \lfloor n/2 \rfloor$ **then**
  6      **return** $q$
  7  **else if** $|L| > \lfloor n/2 \rfloor$ **then**
  8      $G' \leftarrow$ induced subgraph of $G$ on $L$
  9      **return** MEDIANSELECTIONUSINGDAG$(G')$
10  **else**
11      $G' \leftarrow$ induced subgraph of $G$ on $R$
12      $k' \leftarrow \lfloor n/2 \rfloor - |L| - 1$
13      **return** MEDIANSELECTIONUSINGDAG$(G')$ with adjusted rank $k'$

---

## 4.3 Theoretical Framework

We now formalize the key properties and theoretical foundations of the algorithm.

While seemingly definitional, this lemma is crucial as it confirms that a simple graph traversal (BFS) correctly identifies a subset of relationships via transitivity without requiring any new comparisons.

**Lemma 7** (Correctness of BFS Classification). *Let $G = (V, E)$ be a DAG over a set of elements $A$, and let $q \in A$ be a pivot element. Define:*

$$L_q = \{x \in A \mid \text{there exists a path from } x \text{ to } q\}, \quad R_q = \{x \in A \mid \text{there exists a path from } q \text{ to } x\}$$

*Then for all $x \in L_q$, we have $x \prec q$, and for all $x \in R_q$, we have $x \succ q$.*

This classic theorem serves as the cornerstone of our algorithm. It allows us to decompose any complex partial order into a small number ($w$) of simple, totally ordered chains.

**Lemma 8** (Chain Decomposition via Dilworth's Theorem Dilworth [1987]). *Let $G = (V, E)$ be a DAG with width $w$. Then, the vertex set $V$ can be partitioned into at most $w$ disjoint chains $C_1, C_2, \ldots, C_w$, where each $C_i$ is a totally ordered subset of $V$.*

**Lemma 9** (Correctness of DAG-based Partition). *Given a pivot $q$, and chain decomposition $\{C_1, \ldots, C_w\}$, let each chain $C_i$ be ordered, and let $i_q$ be the index of $q$ in its chain $C_i$. Then all elements preceding $q$ in $C_i$ are less than $q$, and all elements following $q$ are greater than $q$, assuming consistency of partial order.*

## 4.4 Complexity Analysis

Let $n$ be the number of elements and $w$ the width of the DAG. The graph traversal, including BFS and inverse BFS, takes $O(n)$ time. Chain decomposition, using known algorithms Cáceres [2023], requires $O(n \log w)$ time. Binary search of $w$ chains has a total number of comparison of $O(w \log(n/w))$ (The reason is given in the proof of Theorem 10). Therefore, the overall number of comparisons is significantly less than $O(n)$ when $w \ll n$, which is typical in structured or low-width datasets.

**Theorem 10** (Comparison Complexity with Structural Information). *Let $G = (V, E)$ be a DAG over $n$ elements with width $w$, and let the median element be selected using Algorithm 3. Then the expected number of comparisons required is $O\left(\log n + w \log\left(\frac{n}{w}\right)\right)$ per recursive step, resulting in total comparison cost $O\left(\left(\log n + w \log\left(\frac{n}{w}\right)\right) \log n\right)$.*

*Moreover, the algorithm adapts naturally to structured domains such as tournament graphs, hierarchical workflows, or ranking-based posets, where the width $w$ is typically small.*

The proofs of Lemmas 7, 8, 9, and Theorem 10 are deferred to the Appendix.

This hybrid approach offers an efficient and principled strategy for median selection in the presence of both noisy comparisons and structured side information. By combining chain decomposition, transitive pruning, and robust refinement, the algorithm achieves correctness while minimizing explicit comparisons—especially in settings with inherent partial order structure.

## 5 Experiments

We empirically evaluate our algorithms in two settings: (1) the classical total order case, where a weak comparison oracle is available, and (2) the partially ordered case, where the input is a synthetic DAG and only standard comparisons are used. In the first setting, our goal is to compute the exact median while minimizing costly strong comparison calls. In the second, we aim to reduce the total number of comparisons by leveraging the partial order structure. All experiments use synthetic data: we vary the weak oracle accuracy in the total-order case and control the DAG width in the partial-order case.

### 5.1 LazySelect with Majority Vote

We begin by evaluating our modified LazySelect algorithm on arrays of size $n = 10^5$ and $n = 10^6$, each a random permutation of $\{0, \ldots, n-1\}$. We report the exact median success rate and the normalized number of strong oracle calls per element across weak oracle accuracies $1 - p \in \{0.51, 0.75, 0.9, 1.0\}$ and weak comparison budgets $c \cdot d$, where $d = \log n$ and $c \in \{1, 2, 4, 8, 16\}$. Each configuration is repeated 100 times, and we report average results.

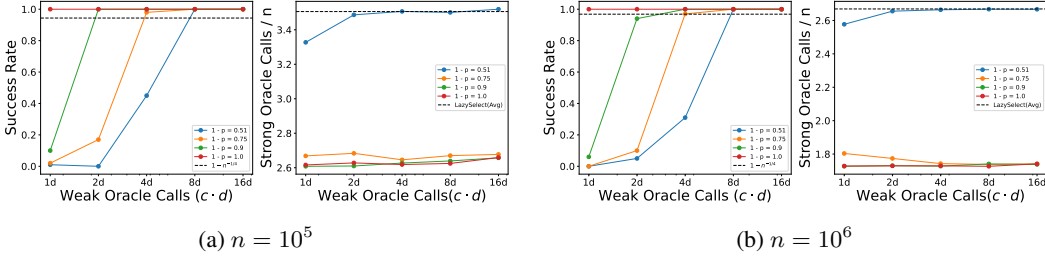

(a) $n = 10^5$          (b) $n = 10^6$

Figure 1: Success rate and the number of strong oracle calls normalized by the array size $n$, with respect to the weak comparison budget $c \cdot d$. Each color represents a different weak oracle accuracy.

e over 94% accuracy when $n = 10^6$. Figure 1 shows that in the noisiest case ($1 - p = 0.51$), only when $c \geq 8$ does the success rate exceed 95%, yet the number of strong comparisons remains nearly identical to the average from 100 runs of standard LazySelect—indicating that, under very noisy conditions, additional weak votes provide negligible benefit. For $1 - p = 0.75$, the success rate exceeds 97% at $c = 4$ and reaches 100% by $c = 8$. Finally, for high-quality oracles ($1 - p \geq 0.9$), only $2 \log n$ weak comparisons suffice to achieve over 94% accuracy. In all cases except when $1 - p = 0.51$, the number of strong oracle calls is lower than that of standard LazySelect. Similar trends are observed for both $n = 10^5$ and $n = 10^6$.

## 5.2 Directed Acyclic Graph Setting

We next evaluate the impact of structural information by comparing both the randomized (Algorithm 3) and deterministic (Algorithm 4) median selection algorithms, with and without access to DAG structure, on synthetic partial orders over $n = 10^4$ elements. DAGs are generated by randomly adding edges while preserving acyclicity, and we control the width $w$ (size of the largest antichain) by varying edge density in steps of 500. Results are averaged over 10 trials per width.

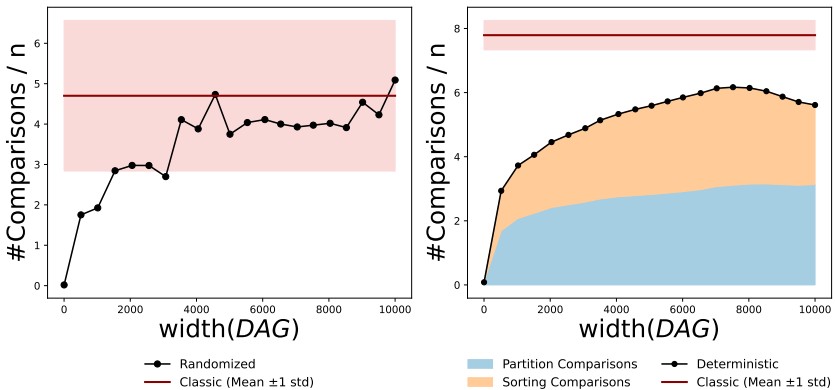

Figure 2: Number of comparisons normalized by the array size $n$, with respect to DAG width $w$. Left: random pivot; Right: median-of-medians pivot. The solid line indicates the performance of the classical median-of-medians algorithm (which ignores the DAG structure), serving as our baseline. Shaded areas show one standard deviation over 1,000 runs for the baseline.

Figure 2 shows that for large widths ($w = 10,000$), both pivot strategies incur approximately 5–6 comparisons (normalized by $n$). As $w$ decreases to around $2,000$, the randomized and deterministic strategies require approximately 3.0 and 4.5 comparisons, respectively. When $w \approx 500$, these values drop to about 1.7 and 2.9. At $w = 1$, both strategies require nearly zero comparisons. The deterministic strategy yields a smoother curve, while the randomized one exhibits more variance but follows a similar downward trend.

**Limitations of Synthetic Data.** Our experiments are conducted on synthetic datasets to enable controlled evaluations where parameters such as noise levels and DAG width can be systematically varied, allowing us to validate our theoretical findings. Evaluating our algorithms on empirical datasets remains an important direction for future work.

## Conclusion

In this paper, we presented a learning-augmented approach to the classical problem of median selection under two distinct yet practically motivated settings. In the first, we explored how to effectively combine weak and strong comparison oracles to identify the median while minimizing the number of expensive strong comparisons. Our algorithm achieves sublinear strong query complexity in expectation, making it well-suited for settings where comparison costs vary significantly. In the second setting, we leveraged partial order information encoded in a DAG to reduce the overall number of comparisons needed for median selection.

To validate our theoretical contributions, we conducted a series of experiments evaluating both the oracle-based and DAG-based algorithms. The results consistently show that our proposed methods outperform standard baselines in terms of comparison efficiency and accuracy.

## Acknowledgments

We thank the anonymous NeurIPS reviewers for their insightful feedback. This work was supported by the New Faculty Startup Fund and conducted with the Algorithmic Foundations of Data Science Lab at Seoul National University.

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

# A Algorithm Deferred from Main Text

**Algorithm 4** Deterministic Median Selection Using DAG (Median of Medians)

**Input:** DAG $G = (V, E)$, where $V$ is a set of $n$ elements
**Output:** Median element of $V$

  1 **if** $n = 1$ **then**
  2     **return** the single element
  3 Divide $V$ into groups of at most 5 elements each
  4 **for** each group **do**
  5     Compute the median of the group using DAG comparisons
  6 Let $M$ be the set of medians from all groups
  7 Construct DAG $G_M$ induced on $M$
  8 $m \leftarrow$ MEDIANSELECTIONUSINGDAG$(G_M)$                 ▷ Recursive call
  9 $L, R \leftarrow$ PARTITIONWITHDAG$(G, m)$
10 **if** $|L| = \lfloor n/2 \rfloor$ **then**
11     **return** $m$
12 **else if** $|L| > \lfloor n/2 \rfloor$ **then**
13     $G' \leftarrow$ induced subgraph of $G$ on $L$
14     **return** MEDIANSELECTIONUSINGDAG$(G')$
15 **else**
16     $G' \leftarrow$ induced subgraph of $G$ on $R$
17     $k' \leftarrow \lfloor n/2 \rfloor - |L| - 1$
18     **return** MEDIANSELECTIONUSINGDAG$(G')$ with adjusted rank $k'$

# B Proof Deferred from Main Text

## B.1 Proof for Lemma 3

*Proof.* **1. Original Middle Region** $M_{\text{original}}$**:**

- The algorithm samples $r = n^{3/4}$ elements uniformly from $A$, sorts them to form $S$, and selects pivots $d$ and $u$ such that:

$$d = S\left[\frac{r}{2} - k\right], \quad u = S\left[\frac{r}{2} + k\right], \quad \text{where } k = \sqrt{n}.$$

- The interval $[d, u]$ in $S$ contains $2k = 2\sqrt{n}$ elements. By the Chernoff bound, the number of elements in $A$ between $d$ and $u$ concentrates around $O(n^{3/4})$ with probability $1 - O(n^{-1/4})$ (see Mitzenmacher and Upfal [2017]).

- Thus, $|M_{\text{original}}| = O(n^{3/4})$ with probability $1 - O(n^{-1/4})$.

**2. Misclassification into $M$:**

- Let $M_{\text{mis}}$ denote elements misclassified into $M$ due to weak comparison errors.

- For an element $a \in L$ (truly $\leq d$), the probability of misclassification is the probability that fewer than $\frac{2}{3} \cdot 2c \log n$ weak comparisons return $<$.

- Let $X$ be the number of correct $\overset{\cdot}{<}$ comparisons. Each comparison succeeds with probability $1 - p$, so:
$$\mathbb{E}[X] = 2c \log n \cdot (1 - p).$$

- Using Hoeffding's inequality for $X \leq \frac{4c}{3} \log n$:

$$\mathbb{P}\left(X \leq \frac{4c}{3} \log n\right) \leq \exp\left(-\frac{2\left(2c \log n \cdot (1 - p) - \frac{4c}{3} \log n\right)^2}{2c \log n}\right).$$

- Simplifying the exponent for $p < 1/2$:

$$\mathbb{P}(\text{misclassification}) \leq \exp(-\Omega(\log n)) = O(n^{-\gamma}),$$

where $\gamma = \Theta(1)$.

## 3. Bounding $|M_{\text{mis}}|$:

- The expected number of misclassified elements is:

$$\mathbb{E}[|M_{\text{mis}}|] \leq n \cdot O(n^{-\gamma}) = O(n^{1-\gamma}).$$

- By Markov's inequality:

$$\mathbb{P}\left(|M_{\text{mis}}| \geq n^{3/4}\right) \leq \frac{\mathbb{E}[|M_{\text{mis}}|]}{n^{3/4}} = O(n^{1-\gamma-3/4}) = O(n^{-1/4}),$$

provided $\gamma \geq 1/4$.

## 4. Total Size of $M$:

- Combining both components:

$$|M| = |M_{\text{original}}| + |M_{\text{mis}}| = O(n^{3/4}) + O(n^{3/4}) = O(n^{3/4}),$$

with probability $1 - O(n^{-1/4})$.

$\square$

## B.2  Proof for Lemma 4

*Proof.* Let $X$ be the number of correct weak comparisons for an element $a \in L$. Each weak comparison succeeds independently with probability $1 - p$, so the expected number of correct outcomes is

$$\mathbb{E}[X] = 2c \log n \cdot (1 - p).$$

For misclassification into the middle region $M$, the number of correct votes must fall below the $\frac{2}{3}$ threshold, i.e., $X \leq \frac{4c}{3} \log n$. Applying Hoeffding's inequality for bounded independent variables:

$$\mathbb{P}\left(X \leq \frac{4c}{3} \log n\right) \leq \exp\left(-\Omega(\log n)\right) = O(n^{-\gamma}),$$

for some constant $\gamma > 0$ that depends on $p$ and $c$.

Summing over all $n$ elements, the expected number of misclassified elements is

$$\mathbb{E}[|M_{\text{mis}}|] = n \cdot O(n^{-\gamma}) = O(n^{1-\gamma}).$$

Applying Markov's inequality, we have

$$\mathbb{P}\left(|M_{\text{mis}}| > O(n^{3/4})\right) = O(n^{-1/4}),$$

for a suitably chosen constant $\gamma > 1/4$. Hence, with high probability, the number of misclassified elements in $M$ is bounded by $O(n^{3/4})$. $\square$

## B.3  Proof for Lemma 5

*Proof.* By Lemma 3, only $O(n^{3/4})$ elements fall into the uncertain middle region $M$, each requiring a constant number of strong comparisons. The number of exact comparison in sorting is $O(n^{3/4} \log n)$ Hence, the total number of strong comparisons is $\tilde{O}(n^{3/4})$ with high probability. $\square$

## B.4  Proof for Lemma 7

*Proof.* By definition of reachability in a DAG, if there is a path $x \to \cdots \to q$, then transitivity of the partial order implies $x < q$. Similarly, if $q \to \cdots \to x$, then $q < x$. $\square$

### B.5 Proof for Lemma 8

*Proof.* This is a direct application of Dilworth's Theorem, which states that the minimum number of chains needed to cover a partially ordered set equals the size of its largest antichain. Since the DAG has width $w$, the size of its largest antichain is $w$, so the vertex set can be decomposed into $w$ chains. $\qquad\square$

### B.6 Proof for Lemma 9

*Proof.* Within each chain $C_i$, the elements are totally ordered. Hence, binary search can be used to find the relative position of $q$, and the local order within each chain ensures the correctness of the partition around $q$. This partitioning respects the partial order and contributes to the global ordering of $A$. $\qquad\square$

## C   Proof of Theorem 10

To determine the relationship between the pivot $q$ and each element $v \in V \setminus \{q\}$, we exploit the structure of the DAG. By Dilworth's Theorem, a DAG of width $w$ can be partitioned into $w$ chains $C_1, C_2, \ldots, C_w$, where each chain is totally ordered.

Let $n_i$ denote the number of elements in chain $C_i$, so that $\sum_{i=1}^{w} n_i = n$. Since elements within a chain are totally ordered, we can determine the relationship between $q$ and any element $v \in C_i$ using binary search within that chain, which takes $O(\log n_i)$ comparisons.

Therefore, the total number of comparisons needed to partition all elements with respect to $q$ is:

$$\sum_{i=1}^{w} O(\log n_i).$$

Applying Jensen's inequality (or noting that the logarithm is concave), we get the bound:

$$\sum_{i=1}^{w} \log n_i \leq w \log \left( \frac{1}{w} \sum_{i=1}^{w} n_i \right) = w \log \left( \frac{n}{w} \right).$$

Thus, the partitioning step takes

$$O(w \log(n/w))$$

comparisons in total.

Including an additional $O(\log n)$ factor for selecting the pivot and bookkeeping, the total expected number of comparisons per recursive step becomes:

$$O \left( \log n + w \log \left( \frac{n}{w} \right) \right).$$

## D   Experiments Setup

Experiments were conducted on a MacBook Air (M1, 2020) with 8 GB RAM. All code was compiled under the C++17 standard with the `-O3` optimization flag. Since our algorithm relies on lightweight comparison operations rather than heavy machine learning training, performance is not significantly affected by hardware. All source code used for these experiments is provided in the supplementary material as a ZIP archive.

