# OpenReview forum: "Median Selection with  Noisy  and Structural Information"
_NeurIPS.cc/2025/Conference — NeurIPS 2025 poster_

### Official Review · Reviewer_YZ2f · 2025-07-01

**Clarity:** 3
**Significance:** 2
**Originality:** 2
**Rating:** 4
**Confidence:** 4

**Summary:**

The paper studies the problem of finding the exact median of a set of $n$ elements under two distinct settings –
1. One has access to a strong (accurate but expensive) oracle along with a weak (inaccurate but inexpensive) oracle.
2. One has access to a DAG i.e. finding median of a partially ordered set.

For the first case, the authors introduce a modified LazySelect algorithm that achieves log-linear running time and sublinear number of calls to the strong oracle. For the latter, they present a recursive partitioning algorithm with its complexity depending on the width of the DAG.

The authors provide theoretical analysis and simulations to support their algorithms.

**Questions:**

1.	How are the two settings related? Can one be smoothly transitioned into another? Would there be any relevance of a joint ‘noisy DAG’ setup?
2.	Can you comment on the generalizability of your algorithms to k-th order statistic?
3.	On line 133, shouldn’t it be $O((\log n)^2)$?
4.	What would be the worst-case input for your algorithms? What input would they fail to return the median on?

**Ethical Concerns:**

["NO or VERY MINOR ethics concerns only"]

**Final Justification:**

A major shortcoming of the paper was the disjointedness of the two settings discussed. The authors have described a unified framework which ties the paper together. Other minor concerns were adequately answered - k-th statistic generalization, comparison to the original lazy select and worst case settings for their algorithms.

The authors also outline the novelty of their approach. While I understand their justification, it was still the main reason why I chose to score the paper at 4 - borderline accept rather than a 5 - accept.

**Quality:**

2

**Strengths And Weaknesses:**

Strengths:
- The paper is well written and organized. The algorithms are clearly described and intuitive, and the theoretical analysis is crisp.
- The context of the problem and prior works are discussed well, both from the perspective of order statistics of data as well as algorithms with predictions paradigm.
- Simulations with synthetic data provide a deeper insight into the performance of the algorithms.

Weaknesses:
- The LazySelect algorithm has been around for a long time and the paper only tweaks it to do an initial filtering based on the weak oracle.
- For the case of DAGs, the Median of Medians or pivot and partition are commonly used algorithms and their combination with a DAG decomposition suggests limited novelty.
- The paper also feels disjoint due to a lack of discussion about the similarity and differences of the two settings.

---

> ### Author Rebuttal · Authors · 2025-07-26
>
> Dear Reviewer YZ2f,
>
> Thank you for your thoughtful feedback and for appreciating the writing, organization, and analysis of our paper. We address your concerns about novelty and the connection between the two settings below.
>
>
>
> **1. On the Novelty of the Algorithms**
>
> We appreciate your perspective on novelty. While our algorithms build on classic paradigms like LazySelect and pivot-and-partition, our key contributions lie in adapting these paradigms to novel computational models with rigorous theoretical guarantees.
>
> In the **noisy oracle setting**, we are the first to integrate a **majority vote mechanism within LazySelect** to overcome persistent noise from a weak oracle. This modification enables us to analyze the interplay between weak and strong oracles and to prove that **a sublinear number of strong queries suffices** for exact median selection. To our knowledge, this is the first formal result of this type.
>
> In the **DAG setting**, our main contribution is connecting the **graph-theoretic structure (DAG width)** to the **comparison complexity** of median selection. This provides a principled way to quantify how much structural side information helps reduce query complexity, yielding the first analysis of median selection in this structured comparison model.
>
> We will revise the introduction and contributions sections to more clearly emphasize that our novelty lies in:
> (i) the theoretical guarantees for these new models, and
> (ii) the novel use of classical tools (e.g., LazySelect with majority voting) in these contexts.
>
>
>
> **2. On the Two Problems Feeling Disjoint**
>
> We greatly appreciate your insightful observation, which has helped us recognize a unifying theme across our two problem settings. The key idea is that the **DAG structure can be interpreted as encoding the outcomes of prior strong oracle queries**. This yields a cohesive framework where both settings aim to minimize the number of costly new queries by leveraging existing comparison information.
>
> **Unified Model:**
>
> * **Prior knowledge representation**:
>
>     – **DAG edges** = outcomes of historical strong oracle queries (trusted comparisons)
>
>     – **Weak oracle** = new noisy comparisons (with persistent errors)
>
>     – **Strong oracle** = new definitive comparisons
>
> We will revise the introduction to explicitly frame this shared motivation and clarify how both settings fit within a unified lens of median selection under side information.
>
>
> **3. On Your Specific Questions**
>
> * **Generalizability to k-th Order Statistic**: Yes, our algorithms can be adapted to find the k-th element by appropriately adjusting the rank thresholds used for pivot selection and final element retrieval.  In the noisy oracle setting, the key idea is to adjust the rank-based pivot selection. The original algorithm selects pivots \$d\$ and \$u\$ to form a confidence interval around the median’s expected rank. To target the *k*-th smallest element, we instead center the interval around the expected rank of the *k*-th element in a random sample \$S\$ of size \$r\$, which is \$r \cdot (k/n)\$. The original success condition, \$|L| < n/2 < |L| + |M|\$, is updated to:
>
>   $$
>   |L| < k \leq |L| + |M|
>   $$
>
>   This ensures that exactly \$k\$ elements lie in the union of \$L\$ (elements less than the target) and \$M\$ (elements near the target). We will add this generalization to the revised manuscript.
>
>
> For the DAG algorithm (Algorithm 3), the recursive step can search for an element of rank $k$ instead of the median ($n/2$). The algorithm already supports $k$-th element selection, and this functionality is implemented in the supplementary document submitted.
>
>
>
> * **Typo on Line 133**: Thank you for pointing this out.
> The correct bound for $w = O(1)$ is indeed $O(\log^2 n)$, as can be seen from Theorem 2, which gives a bound of
> $O\big((\log n + w \cdot \log(n/w)) \cdot \log n\big).$
> However, when the DAG width is exactly 1, there is only a single chain and the recursion depth becomes 1. In this case, we can find the exact median using only $O(\log n)$ comparisons.
> We will revise the corollary accordingly. The main theorem remains correct as stated.
>
> * **Worst-Case Inputs and Failure Modes**:
>
>    • **Algorithm 1 (Noisy setting)**: The worst case occurs when the random sample $S$ is unrepresentative, leading to poor pivots $d$ and $u$. Combined with adversarial noise from the weak oracle, this can (with low probability) cause the true median to fall outside of the final narrowed region.
>
>    • **Algorithm 2 (DAG setting)**: The worst case is when the DAG has large width $w \approx n$, offering little help over classical methods. In this case, our algorithm matches the standard comparison complexity up to constants.
>
> We will include a brief discussion of these cases in the revision.
>
>
> Thanks again for your detailed review and constructive suggestions.

---

> > ### Comment · Reviewer_YZ2f · 2025-08-06
> >
> > The authors have adequately answered my questions. I will revise my rating accordingly.
> >
> > Based on the feedback from all four reviewers, the authors have promised several additions to their paper and these will be crucial for completeness.

---

### Official Review · Reviewer_2hww · 2025-07-02

**Clarity:** 4
**Significance:** 3
**Originality:** 3
**Rating:** 4
**Confidence:** 4

**Summary:**

This paper studies the median selection problem with noisy comparisons results. In this paper's setting, each pair of items can only be queried once (further queries will give the same results). When querying a strong oracle, the correct relations of them will be returned. When querying a weak oracle, the correct relation will be returned with probability 1-p>1/2. This paper further studies the case with a prior known comparison relations (represented by a DAG).

For the first case, this paper proposes an algorithm that needs O(n\logn) weak oracle queries and \tilde{O(n^0.75)} queries of the strong oracle, and returns the correct median with probability at least 1 - O(n^{-1/4}). For the structural case, this paper proposes an algorithm with O((\log(n) + w\log(n/w) log(n)) query complexity, where w is the width of the DAG.

**Questions:**

-Any previous works studying top-1 selection or k-th item selection? What's the differences between them and the median selection?
-If we can use more or less strong oracle queries, how many numbers of weak oracle queries will be needed?

**Ethical Concerns:**

["NO or VERY MINOR ethics concerns only"]

**Limitations:**

No negative social or ethnic concerns

**Paper Formatting Concerns:**

No formatting concerns

**Quality:**

3

**Strengths And Weaknesses:**

Strengths
-The problem studies in this paper is very interesting and foundational. In a lot of real-world scenarios, querying the same pair multiple times is not possible. This setting differs from the dueling bandit settings, where multiple queries of the same pair can get more and more confidence relations.
-The algorithms and theoretical analysis are clean and sound. The results look good from my intuition.

Weakness
-The two problems in this paper do not look very close. Maybe put them into two papers and enrich each paper with more contents and results?
-The results lack baselines to evaluate. Either previous works or lower bounds can be helpful here.

---

> ### Author Rebuttal · Authors · 2025-07-26
>
> **Dear Reviewer 2hww,**
>
> Thank you for your very positive and encouraging review. We are glad that you found the problem interesting and our analysis clean and sound. Below, we address your specific points in detail.
>
>
>
> **1. On the Two Problems Feeling Disjoint**
>
> We greatly appreciate your insightful observation, which has helped us recognize a unifying theme across our two problem settings. The key idea is that the *DAG structure can be interpreted as encoding the outcomes of prior strong oracle queries*. This perspective yields a cohesive framework where both settings aim to leverage existing comparison information to minimize the use of costly new queries:
>
> * **Unified Model**
>
>   * *Prior Knowledge Representation*:
>
>     * **DAG edges** = outcomes of historical strong oracle queries (trusted comparisons)
>     * **Weak oracle** = new noisy comparisons (with persistent errors)
>     * **Strong oracle** = new definitive comparisons
>
> We will add a paragraph in the introduction to explicitly frame this shared motivation and highlight the connection between the two parts of the paper under this unified lens.
>
>
> **2. On the Lack of Baselines and Lower Bounds**
>
> This is a very important point.
>
> * **Noisy Oracle Setting**:
>   To the best of our knowledge, our work is the first to consider the combination of strong and weak oracles for *exact* median selection. This makes it challenging to find a direct baseline from prior work. However, we agree that discussing theoretical lower bounds would strengthen the paper. One natural lower bound is that *it is not possible to achieve exact selection using only \$O(n)\$ weak queries and \$o(n)\$ strong queries*—a fact that highlights the necessity of our hybrid design. We will include this discussion in the revision.
>
> * **DAG Setting**:
>   We apologize for not making the baseline clearer. The "Classic (Mean ±1 std)" lines in Figure 2 correspond to the performance of the standard median-of-medians algorithm *without access to DAG information*. We will revise the figure captions and clarify this in the main text.
>
>
> **3. Responses to Your Specific Questions**
>
> * **Generalizing to *k*-th Item Selection**:
>   Yes, our algorithm naturally generalizes. In the noisy oracle setting, the key idea is to adjust the rank-based pivot selection. The original algorithm selects pivots \$d\$ and \$u\$ to form a confidence interval around the median’s expected rank. To target the *k*-th smallest element, we instead center the interval around the expected rank of the *k*-th element in a random sample \$S\$ of size \$r\$, which is \$r \cdot (k/n)\$. The original success condition, \$|L| < n/2 < |L| + |M|\$, is updated to:
>
>   $$
>   |L| < k \leq |L| + |M|
>   $$
>
>   This ensures that exactly \$k\$ elements lie in the union of \$L\$ (elements less than the target) and \$M\$ (elements near the target). We will add this generalization to the revised manuscript.
>
>
> For the DAG algorithm (Algorithm 3), the recursive step can search for an element of rank $k$ instead of the median ($n/2$). The algorithm already supports $k$-th element selection, and this functionality is implemented in the supplementary document submitted.
>
>
>
> * **Trade-off Between Strong and Weak Oracles:**
> This is an excellent and thought-provoking question. The strong oracle is primarily used in two places: (1) to sort the elements in the sample set $R$, and (2) to order the elements in the middle region $M$, i.e., those between the pivots $d$ and $u$. There is an inherent trade-off here: sampling more elements into $R$ reduces the size of $M$, thereby requiring fewer strong oracle comparisons in $M$; conversely, sampling fewer elements increases the burden on the strong oracle to sort a larger $M$. We believe that choosing $\Theta(\sqrt{n}poly\log n)$ elements strikes a balance between these two costs. Consequently, we argue that $\Theta(\sqrt{n})$ strong oracle queries are both necessary and sufficient. We will clarify this intuition and include this remark in the revised version.
>
>
> Thank you again for your detailed and highly valuable feedback.

---

### Official Review · Reviewer_NEam · 2025-07-02

**Clarity:** 2
**Significance:** 3
**Originality:** 2
**Rating:** 4
**Confidence:** 4

**Summary:**

This paper studies the problem of finding a median from 1) a complete order using noisy pairwise comparison queries and 2) a partial order using non-noisy queries.

For the first problem, they are given an order over n elements. They are allowed to make two types of queries over pairs of points: 1) a strong query, which always returns the correct order between the points, and 2) a weak query, which outputs an incorrect order with probability < 0.5. A good solution should use sublinear strong queries, and at most linear (preferably sublinear) weak queries. This model is based off a handful of previous works in related areas. The problem has not been studied with respect to strong and weak queries together, but it has been studied with respect to strong queries only.

Their solution is an adaptation of Rivest and Floyd's lazy select algorithm, which is loosely modeled after quicksort. What they do is they sample a large (but sublinear) set of points as a sort of "pivot set" akin to the quicksort algorithm. With high probability, the middle 2sqrt(n) of sampled points encapsulates the median. They take points along the edge of this middle range and use them to sort the rest of the points as low, high, or in the middle range. When they sort a single point, they compare them to all sampled points in this range, and if sufficiently many state it is low or high, then it is sorted as such. If there is not a strong enough majority, then strong comparisons are used to definitively order the points. After sorting, the middle range is ordered and they can identify the median from the middle range with high probbaility.

For the partially ordered problem, they use existing methods to decompose the DAG representing the order into a number of disjoint chains, and then they can use binary search methods along these chains to implement a sort of quicksort algorithm. Note that this problem is generally easier than the other problem, since a median is only defined by having sufficiently few smaller or larger nodes than this (and so having more unorder in the partial order means more points could constitute a valid median).

Experiments are run on synthetic datasets. For the completely ordered problem, random permutations are used. They find that using logarithmically many weak oracle calls, they can achieve near perfect success rate in practice, but if the probability of an incorrect label is too high, the number of strong oracle calls is also high. For the partially ordered problem, DAG are generated by random edge generation, and they see that the number of comparisons decrease as the DAG width decreases.

**Questions:**

1. In Theorem 2, it doesn't seem like the "in particular" part follows. The general total comparisons you have can be rewritten as O(log^2n + w log(n/w) log(n)). Therefore, no matter what w is, it can't be below O(log^2 n). Additionally, if w = n / log(n) = o(n), the latter term becomes O(n loglog(n)), which is not o(n) as stated. Perhaps these are additional results you found that aren't important special cases of the general formula you have above? If so please clarify that, because it's confusing as written.

2. If the error of the weak oracle were not fixed, would you even be able to leverage resampling to obtain a better algorithm? As in, is this assumption necessary to make your results interesting?

3. Do you think your work has any notable relation to rank aggregation? In this problem, you're given a number of permutations over [n] and are asked to output a sort of "median ranking" over the set. Quicksort has a nice analog here, where you create your ordering using pivots and then taking a majority vote over the permutations asking if each value is higher or lower than the pivot. The majority vote is nontransitive, meaning there is uncertainty of what order would result dependent on the order of pivot selection. This is obviously distinctly different from the noise you're considering, but it's reminiscent of it, and I'm wondering if some techniques are transferable between the problems. (This is also similar to the DAG formulation, but here we may have cycles).

4. In Section 3.1, the algorithm description, step 5, what are you comparing a to for the strong oracle? Also you did not adequately describe what criteria puts a into M.

5. For the DAG problem, you discuss "the true median" on line 243. Aren't there potentially many possible medians that satisfy this for a DAG? I mean, theoretically every point could be a median. The way you word it here seems to imply it's unique.

6. Are Lemmas 7 and 9 not just restating the definition of the order imposed by the DAG?

7. Please include a brief definition of DAG width.

8. What do you mean by "the algorithm adapts naturally to structured domains such as..."?

**Ethical Concerns:**

["NO or VERY MINOR ethics concerns only"]

**Final Justification:**

This work is solid but I do not think the writeup is ideal quality for an accepted paper.

**Limitations:**

Yes

**Quality:**

3

**Strengths And Weaknesses:**

The median selection problem is a fundamental problem so any work on this problem is interesting. The noisy model they use has been used before and makes some intuitive sense, though I would really like to see more evidence that it is a useful model to look at. Their results for both problems are sensible and imply nice, straightforward algorithms. The methods are somewhat interesting extensions of existing algorithms.

The writing is easy to follow but they admit most of the intuition as to why their methods work. Their analysis sections in 3.2, 4.3, and 4.4 are basically lists of lemmas which I find nearly unacceptable. The experiments are nice to have and portray interesting trends, but I would say their interest is limited partly by how little experimental analysis was done but also because the synthetic dataset generation is extremely basic and not likely to resemble real world distributions.

---

> ### Author Rebuttal · Authors · 2025-07-26
>
> **Dear Reviewer NEam,**
>
> Thank you for your thorough and thoughtful review. We appreciate your positive remarks on the importance of the problem and the clarity of our algorithmic contributions. Below, we address your concerns and questions point by point.
>
>
>
>  1. **On the Presentation of the Analysis**
>
> We agree that presenting the analysis as a "list of lemmas" without sufficient intuition is not ideal. While the complete formal proofs are provided in the appendix, we will revise the main text to include high-level proof sketches and intuitive explanations for each lemma to improve readability and motivation.
>
>
>
> 2. **On the Synthetic Nature of Experiments**
>
> We acknowledge that our experiments are based on synthetic data. This was an intentional design choice to create a controlled setting where we can precisely vary key parameters such as noise levels and DAG widths to empirically validate our theoretical results. We will add a discussion in the limitations section to acknowledge that real-world data may introduce additional complexity and that evaluating the algorithm on such datasets is an important direction for future work.
>
>
>  3. **Responses to Specific Questions**
>
> * **Theorem 2 Complexity:**
>   Thank you for catching this.
> The correct bound for $w = O(1)$ is indeed $O(\log^2 n)$, as can be seen from Theorem 2, which gives a bound of
> $O\big((\log n + w \cdot \log(n/w)) \cdot \log n\big).$
> However, when the DAG width is exactly 1, there is only a single chain and the recursion depth becomes 1. In this case, we can find the exact median using only $O(\log n)$ comparisons.
> We will revise the corollary accordingly. The main theorem remains correct as stated.
>
> * **Persistent Weak Oracle Error:**
>   The assumption of persistent noise is standard in the literature and essential to the non-triviality of the model. Without persistence, one could simply resample the same pair $O(\log n)$ times to reduce error probability to $1/\text{poly}(n)$, rendering the weak oracle nearly as powerful as a strong one. The persistent model captures systematic noise, such as a biased annotator. We will add this clarification to the revised version.
>
> * **Relation to Rank Aggregation:**
>   This is an excellent observation. The main distinction lies in the assumption of an underlying total order in our setting, which ensures transitivity—unlike in rank aggregation problems, where preference cycles (e.g., Condorcet paradoxes) may occur. That said, both models aim to infer global rankings from noisy input, and we will add a brief note in the related work section to highlight this connection.
>
> * **Algorithm 1, Step 5 (Use of Strong Oracle):**
>   We apologize for the ambiguity. In Step 5, the strong oracle is used to compare an element $a$ against the pivots $d$ and $u$, determining definitively whether $a < d$, $d \leq a \leq u$, or $a > u$. An element is placed in the middle set $M$ if it cannot be confidently placed in $L$ or $R$ using the weak oracle or if it is explicitly placed there by the strong oracle. We will revise the description of Algorithm 1 to make this process more explicit.
>
> * **"True Median" Terminology:**
>   You're right—the phrase “the true median” is misleading in the context of a DAG. What we mean is the median with respect to some total order consistent with the DAG. We will revise this phrasing to “a median of a consistent total order” to ensure precision.
>
> * **Lemmas 7 and 9:**
>   While these lemmas may appear definitional, they are essential steps in our correctness analysis. They formally verify that our BFS traversal and chain-based partitioning procedures correctly leverage the transitive structure of the DAG to classify elements without error.
>
> * **Definition of DAG Width:**
>   We will add a formal definition of DAG width (i.e., the size of the largest antichain) in the preliminaries to eliminate ambiguity.
>
> * **"Adapts naturally to structured domains...":**
>   We agree that this statement can be more concrete. We will revise it to:
>   *“The algorithm’s performance, as shown in Theorem 2, improves as the width $w$ of the DAG—representing its structural complexity—decreases.”*
>
>
> Thank you again for your detailed and highly valuable feedback.

---

> ### Comment · Reviewer_NEam · 2025-08-08
>
> I am mostly satisfied by the authors' responses. Still, given the quality of the submission writeup as given, I am not going to increase my score. I think it would be reasonable to accept this paper given the authors' commitments to revision, and otherwise I would be happy to see the revised paper in upcoming submissions.

---

### Official Review · Reviewer_JGL7 · 2025-07-03

**Clarity:** 2
**Significance:** 2
**Originality:** 2
**Rating:** 4
**Confidence:** 4

**Summary:**

The paper studies exact median selection under two settings: (1) noisy comparisons
 from both weak (inexpensive, error-prone) and strong (accurate, costly)
oracles, and (2) inputs with partial order represented as DAGs.

For the first setting, the authors propose a modiffed LazySelect algorithm that uses majority
 voting among weak comparisons to reduce the number of strong oracle calls,
achieving sublinear strong comparison complexity with high-probability correctness.

For the second setting, the authors exploit structural information in the
DAG to design a median selection algorithm that requires signiffcantly fewer
comparisons when the DAG has small width.

Both algorithms are analyzed theoretically and supported by experiments on synthetic data.

**Questions:**

1. Can a similar method be used to ffnd the kth largest number instead of
the median?

2. Is it possible to keep the weak oracle query at O(n) in Theorem 1?

**Ethical Concerns:**

["NO or VERY MINOR ethics concerns only"]

**Final Justification:**

The authors have clarified most of my questions and concerns, including (1) the clarification of the algorithm's generalizability to finding the k-th largest element, (2) the explanation of the theoretical impossibility of reducing weak oracle queries, and (3) the clarification of this work's main contribution. Overall, the theoretical analysis of the paper is solid. However, the core approach that using majority voting to replace one strong query with O(log n) weak queries is somewhat straightforward, hence my final rating of borderline accept.

**Limitations:**

The algorithm for solving noisy comparison will call weak oracle O(n log n)
times. The algorithm for solving DAG improves only when the width is small,
and performs worse than the classic algorithm when the width is close to n.

**Quality:**

2

**Strengths And Weaknesses:**

Strengths
1. The paper effectively integrates weak and strong comparisons, achieving
sublinear use of the expensive strong oracle.
2. By leveraging partial order (DAG) structure, the proposed algorithm signiffcantly
 reduces comparison complexity when the DAG has small width.
3. The experiments validate both algorithms under varied noise levels and
DAG structures, conffrming the theory.

Weaknesses
1. Theorem 1 shows that the total number of oracle calls of the proposed
algorithm is O(n log n), which is larger compared to the O(n) of median
of median algorithm or lazy selection algorithm mentioned in the paper.
Since the probability of a weak oracle being correct is greater than 1/2,
it is not a very nontrivial result to be able to use log n times of the weak
oracle instead of once the strong oracle.
2. While Algorithm 1 presents an interesting modiffcation of the LazySelect
approach, the paper would benefft from a more explicit comparison with
the original LazySelect algorithm.
3. According to the conclusion of Theorem 2 recounted at line 131, the corollary
 after line 132 is incorrect, with one less log n term taken into account.

---

> ### Author Rebuttal · Authors · 2025-07-26
>
> **Dear Reviewer JGL7,**
>
> Thank you for your valuable time and constructive feedback on our work. We appreciate that you found our integration of weak/strong oracles and the use of DAG structures effective. Below, we address your specific concerns in detail.
>
>
>  **Addressing Questions**
>
> #### **1. Generalization to k-th Largest Element**
>
> Yes, Algorithm 1 can be generalized to compute the *k*-th largest element, not just the median. The core idea—using a sample to identify pivot elements that isolate the target in a smaller subproblem—remains unchanged. The required modifications are as follows:
>
> ##### 1.1 Adjusting the Pivot Selection
>
> The original algorithm selects pivots $d$ and $u$ to form a confidence interval around the median’s expected rank. To generalize this to the *k*-th element, we center the interval around the expected rank of the *k*-th element in the random sample $S$ of size $r$, which is $r \cdot (k/n)$.
>
> **Modification:** The bounds $d$ and $u$ are redefined as:
>
> $$
> d = S\left[\left\lfloor r \cdot \left(\frac{k}{n}\right) - c_k \right\rfloor\right], \quad
> u = S\left[\left\lceil r \cdot \left(\frac{k}{n}\right) + c_k \right\rceil\right]
> $$
>
> This adjustment ensures that the selected range accurately captures the desired rank interval around the $\frac{k}{n}$-quantile, scaled by the sample size $r$ and buffered by $c_k$.
>
> Here, $c_k$ is a cushion parameter (analogous to the original setting $c_k = \sqrt{n}$) that guarantees the true *k*-th element lies between $d$ and $u$ with high probability.
>
> ##### 1.2 Updating the Success Condition and Final Selection
>
> The original condition for the median, $|L| < n/2 < |L| + |M|$, must be updated for the *k*-th rank.
>
> **Modification:** The new success condition becomes:
>
> $$
> |L| < k \leq |L| + |M|
> $$
>
> This ensures that exactly $k$ elements lie in the union of $L$ (less than target) and $M$ (equal to or around the target).
>
> **Modification:** Once this condition is satisfied, the algorithm sorts the set $M$ and returns the element at position $k - |L|$ in the sorted list.
>
> With these updates, Algorithm 1 generalizes to compute any order statistic. Theoretical guarantees, such as achieving sublinear strong oracle complexity with high probability, still hold under this modification.
>
>
>
> For the DAG algorithm (Algorithm 3), the recursive step can search for an element of rank $k$ instead of the median ($n/2$). The algorithm already supports $k$-th element selection, and this functionality is implemented in the supplementary document submitted.
>
>
>
>
> #### **2. Reducing Weak Oracle Queries to $O(n)$**
>
> We agree this is a fundamental challenge. In fact, we can formally argue that no algorithm can achieve high-probability median selection with only
> $O(n)$ *weak queries* (with correctness probability $p = \frac{1}{2} + \epsilon$) and
> $o(n)$ *strong queries*, due to two core limitations:
>
> * **Weak Oracle Limitation:**
>   Previous work showed that using only $O(n)$ weak queries leaves each element misclassified with constant probability, leading to $\Theta(n)$ uncorrected errors.
>
> * **Strong Oracle Sparsity:**
>   With only $o(n)$ strong queries, we can correct at most $o(n)$ errors. Thus:
>
>   $$
>   \Theta(n) - o(n) = \Theta(n)
>   $$
>
>   errors remain uncorrected, causing median misidentification with probability $1 - o(1)$.
>
> The $O(n \log n)$ weak query complexity in our algorithm arises from the majority-vote subroutine, which requires $O(\log n)$ weak comparisons per element to amplify correctness. However, when the oracle’s error rate is sufficiently low (e.g., $p = 1/2 + \Omega(1/\log n)$), the number of required weak comparisons per element can drop to $O(1)$. We will explore and clarify this trade-off in the revised version.
>
>
>  **Addressing Weaknesses**
>
> #### **Weakness #2: Comparison with Original LazySelect**
>
> We agree that a direct comparison with the original LazySelect will enhance the paper. In the classical LazySelect, all comparisons are strong and error-free, with total query complexity $O(n)$. Our variant replaces most strong comparisons with $O(\log n)$ weak comparisons per element (via majority voting), while reducing strong queries to $\tilde{O}(n^{3/4})$. Though total oracle calls rise to $O(n \log n)$, this is a favorable trade-off given the high cost of strong queries. We will add a comparison table to highlight this trade-off in the revised manuscript.
>
> #### **Weakness #3: Theorem 2 Corollary**
>
> Thank you for pointing out the imprecise phrasing.
> The correct bound for $w = O(1)$ is indeed $O(\log^2 n)$, as can be seen from Theorem 2, which gives a bound of
> $O\big((\log n + w \cdot \log(n/w)) \cdot \log n\big).$
> However, when the DAG width is exactly 1, there is only a single chain and the recursion depth becomes 1. In this case, we can find the exact median using only $O(\log n)$ comparisons.
> We will revise the corollary accordingly. The main theorem remains correct as stated.
>
>
> #### **Weakness #1: Nontriviality of Sublinear Strong Queries**
>
> While the weak oracle is more powerful than random guessing, our key contribution lies in the hybrid design: leveraging weak queries while preserving sublinear use of expensive strong queries (i.e., $\tilde{O}(n^{3/4})$). Prior works typically assume either all-clean or all-noisy settings. Our result bridges these extremes. We will clarify this point in the revision to better highlight its novelty.
>
>
>  **Addressing Limitations**
>
> #### **Noisy Comparison Algorithm’s Weak Query Usage**
>
> The use of $O(n \log n)$ weak queries is necessary for robustness via majority voting. While this exceeds the complexity of noiseless algorithms, it is a necessary overhead for dealing with unreliable comparisons. In future revisions, we will discuss possible improvements, such as adaptive voting strategies that reduce overhead when the oracle’s bias is large.
>
> #### **DAG Algorithm’s Dependence on Width**
>
> As noted, the performance of our DAG-based algorithm depends critically on the width $w$. This is unavoidable: when $w = \Theta(n)$ (e.g., in a complete DAG), no structural advantage exists and $\Omega(n)$ comparisons are needed. We will emphasize this structural trade-off more clearly in the final version.
>
> Thank you again for your detailed and highly valuable feedback.

---

> > ### Comment · Reviewer_JGL7 · 2025-08-03
> >
> > The authors have clarified most of my questions and concerns, including (1) the clarification of the algorithm's generalizability to finding the k-th largest element, (2) the explanation of the theoretical impossibility of reducing weak oracle queries, and (3) the clarification of this work's main contribution .

---

### Decision · Program_Chairs · 2025-09-17

**Decision:**

Accept (poster)

**Comment:**

The paper provides algorithms for median estimation using weak and strong comparison oracles. In the classical setting where no order information is provided, they give an algorithm using quasilinear weak comparisons and sublinear strong comparisons. In the presence of additional partial order information, they obtain better results depending on the partial order "width".

Overall, reviewers found the problem and solutions natural, and were happy with the provided guarantees. Several reviewers raised many clarifying questions, and while they were largely satisfied by the author responses, their ratings were contingent on the authors adding these clarifications to the submission. The authors should make sure to reread the discussion and edit the final version accordingly.